# Enhanced copper anticorrosion from Janus-doped bilayer graphene

Mengze Zhao[1,13], Zhibin Zhang [1,2,13] ✉, Wujun Shi [3,4,13], Yiwei Li[5,6], Chaowu Xue[5], Yuxiong Hu[5], Mingchao Ding [7], Zhiqun Zhang[5], Zhi Liu [5,12], Ying Fu[2], Can Liu [8], Muhong Wu [2,9,10], Zhongkai Liu [5], Xin-Zheng Li[1,10,11], Zhu-Jun Wang [5] ✉ & Kaihui Liu [1,2,9,11] ✉

The atomic-thick anticorrosion coating for copper (Cu) electrodes is essential for the miniaturisation in the semiconductor industry. Graphene has long been expected to be the ultimate anticorrosion material, however, its real anticorrosion performance is still under great controversy. Specifically, strong electronic couplings can limit the interfacial diffusion of corrosive molecules, whereas they can also promote the surficial galvanic corrosion. Here, we report the enhanced anticorrosion for Cu simply via a bilayer graphene coating, which provides protection for more than 5 years at room temperature and 1000 h at 200 °C. Such excellent anticorrosion is attributed to a nontrivial Janus-doping effect in bilayer graphene, where the heavily doped bottom layer forms a strong interaction with Cu to limit the interfacial diffusion, while the nearly charge neutral top layer behaves inertly to alleviate the galvanic corrosion. Our study will likely expand the application scenarios of Cu under various extreme operating conditions.

Cu is widely used in the modern semiconductor industry as electrical connectors due to its high conductivity[1–5]. However, the susceptibility of Cu to oxidation greatly limits its advanced applications[6]. For example, Cu will lose more than 99.99999% of its electrical conductivity at 200 °C in air for only 30 min[7]. Conventional organic coating[8] and sacrificial anode[9] anticorrosion techniques have made significant success in traditional industries, unfortunately, these methods typically rely on a thick coating and fail to comply with the miniaturisation requirements of the integrated circuits industry[1,2]. Currently, the rise of two-dimensional

(2D) materials has provided new opportunities for the development of anticorrosion technology[10–12]. Graphene, in particular, has emerged as a promising candidate for an ultimate anticorrosion coating due to its excellent characteristics at the atomic thickness, including absolute impermeability, high electrical and thermal conductivity, excellent chemical stability, and great optical transparency[13–15]. However, previous studies have shown that the anticorrosion performance of graphene is far from satisfactory under real working conditions, especially in complex and diverse environments[16–29], which limits its application to an early stage.

[1]State Key Laboratory for Mesoscopic Physics, Frontiers Science Centre for Nano-optoelectronics, School of Physics, Peking University, Beijing, China. [2]Songshan Lake Materials Laboratory, Institute of Physics, Chinese Academy of Sciences, Dongguan, China. [3]Center for Transformative Science, ShanghaiTech University, Shanghai, China. [4]Shanghai High Repetition Rate XFEL and Extreme Light Facility (SHINE), ShanghaiTech University, Shanghai, China. [5]School of Physical Science and Technology, ShanghaiTech University, Shanghai, China. [6]Institute for Advanced Studies (IAS), Wuhan University, Wuhan, China. [7]Beijing National Laboratory for Condensed Matter Physics, Institute of Physics, Chinese Academy of Sciences, Beijing, China. [8]Department of Physics, Renmin University of China, Beijing, China. [9]International Centre for Quantum Materials, Collaborative Innovation Centre of Quantum Matter, Peking University, Beijing, China. [10]Interdisciplinary Institute of Light-Element Quantum Materials and Research Centre for Light-Element Advanced Materials, Peking University, Beijing, China. [11]Peking University Yangtze Delta Institute of Optoelectronics, Nantong, Jiangsu, China. [12]Present address: Center for Transformative Science, ShanghaiTech University, Shanghai, China. [13]These authors contributed equally: Mengze Zhao, Zhibin Zhang, Wujun Shi. ✉e-mail: zhibinzhang@pku.edu.cn; wangzhj3@shanghaitech.edu.cn; khliu@pku.edu.cn

In principle, an ideal anticorrosion coating material should comply with three requirements: (i) intrinsically, being impermeable to corrosive agents and having low penetrating defect density; (ii) interfacially, being tightly coupled with metal to prevent further horizontal diffusion once corrosive agents reach the interface, and (iii) surficially, being electrically inert to prevent electrochemically facilitated corrosion reactions. A defect-free graphene film naturally fulfils the first requirement, but it generally fails to meet the interfacial and surficial ones. First, the easily formed wrinkles on graphene and intrinsic step bunches on Cu allow for the diffusion of corrosive molecules at the interface[13,22,23]. Second, graphene is cathodic to most metals and will lead to local galvanic reaction[21,24]. As a result, corrosive molecules will interact with the free electrons on graphene surface and thus accelerate the electrochemical corrosion process[25–28]. Recently, efforts have been made to enhance the graphene-Cu couplings to prevent interfacial molecular diffusion, for example, by forming a commensurate graphene/Cu(111) system[15,30] or producing an ultra-flat Cu surface[13]. However, these methods impose overly restrictive limitations on Cu or graphene and still cannot address the electrochemical corrosion issue. Therefore, there is an urgent need to develop a robust and effective anticorrosion strategy for graphene coating that can address all those three requirements.

Here, we utilized a bilayer graphene coating and investigated its enhanced anticorrosion performance for Cu[17–20,31]. It is found that the bilayer coating strategy not only integrates the advantages of the monolayer (impermeability, stability, and optical transparency), but also demonstrates excellent prospects far beyond expectation (with a Janus-doping effect for improved anticorrosion). Specifically, it can simultaneously satisfy the demands of both reinforcing the graphene-Cu interactions to limit interfacial molecular diffusion and maintaining the intrinsic charge neutrality of the top layer graphene to suppress surficial electrochemical reactions (Fig. 1a).

## Results and discussion

In our experiments, we first obtained a large (~7 × 7 cm$^2$) arbitrary-twisted bilayer graphene coating on Cu by wet transfer method (see Methods for more details). After being oxidized at 250 °C for 6 hours, the monolayer graphene coated Cu quickly deteriorated and turned black (the colour of Cu oxide, Cu$_x$O), whereas the bilayer coated area maintained the initial Cu state with shining metallic lustre (Fig. 1b), demonstrating an enhanced anticorrosion performance. We also tracked a Cu foil with directly grown bilayer domains for a long-term storage, and the results showed that bilayer graphene could effectively protect the underneath Cu from oxidation for more than five years (Fig. 1c). In contrast, the monolayer graphene coated Cu had already been oxidized with an obvious Cu$_x$O layer, as shown in the optical images and the Raman spectra (Fig. 1c, d).

Cross-sectional scanning transmission electron microscopy (STEM) images further confirmed that the Cu$_x$O layer only formed in the monolayer graphene coated region, while no oxide was observed in the bilayer coated area, even on the bunched Cu steps (Fig. 1e and Supplementary Fig. 1). Previous studies have shown that the oxidation process typically involves stepped surfaces, where corrosive molecules might diffuse along the bunched Cu steps to form oxides[13,32]. However, our results demonstrated that even on a highly stepped high-index surface, the bilayer graphene coating could still effectively protect Cu (Supplementary Fig. 2). More interestingly, the anticorrosion performance of the bilayer coating is found to be independent of the twist angle (Supplementary Fig. 3), regardless of whether the twisting was introduced by the direct growth or the transfer process (see Methods for more details). Furthermore, to investigate the anticorrosion performance of bilayer graphene in the presence of water, we performed wet-oxidation tests at 150 °C with an environment of 70% relative humidity and at 80 °C with the sample immersed in water (Supplementary Fig. 4). Both experiments demonstrated the enhanced anticorrosion properties of bilayer graphene to monolayer ones.

To quantitatively evaluate the improved Cu anticorrosion performance of bilayer graphene, we continually monitored and recorded the corrosion process. After exposure to air at 200 °C, monolayer graphene coated Cu gradually oxidized, and more than 70% area of Cu deteriorated after 15 days. In contrast, bilayer graphene coated Cu barely oxidized throughout the entire oxidation period (Fig. 2a and Supplementary Fig. 5). We further constructed a two-step defect formation and expansion model[33,34] (see Methods for more details), and deduced from the model that bilayer graphene could effectively

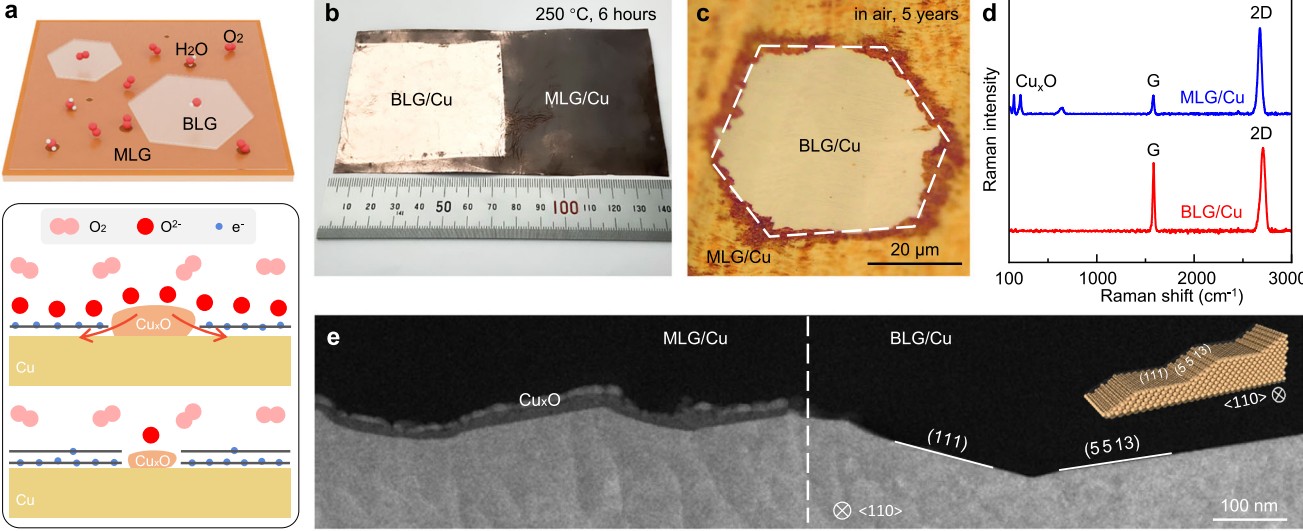

**Fig. 1 | Enhanced Cu anticorrosion performance of bilayer graphene.**
**a** Schematic for the oxidation process of monolayer graphene (MLG) and bilayer graphene (BLG) coated Cu foils. The top panel shows the three-dimensional view and the bottom panel shows the side view. **b** Photograph of the monolayer and bilayer graphene coated Cu after being oxidized at 250 °C for 6 h. **c** Optical image of the monolayer and bilayer graphene coated Cu after storage at room temperature for 5 years in Beijing, China. **d** Raman spectra corresponding to the monolayer and bilayer graphene coated area in **c**. **e** Cross-sectional STEM image of the monolayer (left) and bilayer (right) graphene coated Cu after being oxidized at 270 °C under 230 Pa O$_2$ for 10 min. Inset: atomic stacking model of Cu showing the crystal plane of the steps.

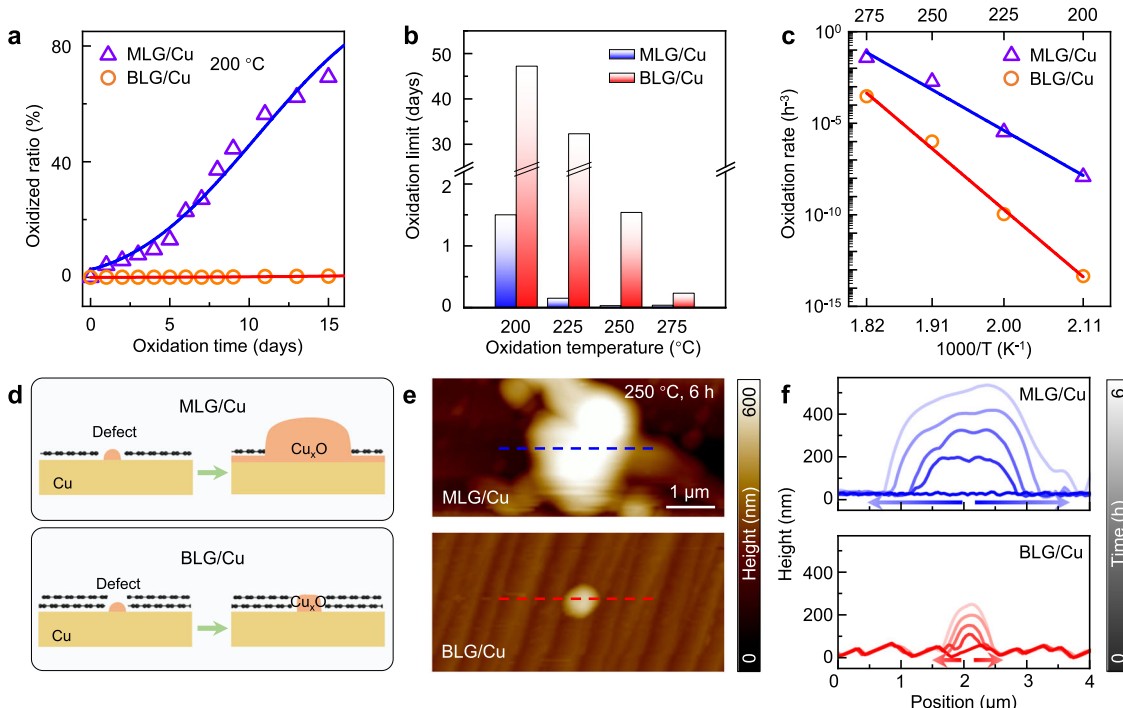

**Fig. 2 | Cu corrosion resistance of monolayer and bilayer graphene. a** Evolution of the oxidized ratio of monolayer (triangle) and bilayer (circle) graphene coated Cu when heated at 200 °C in air. **b** The oxidation limit (5% area of the Cu was oxidized) of monolayer and bilayer graphene coated Cu at different temperatures. **c** Arrhenius plot of the oxidation rates of monolayer (triangle) and bilayer (circle) graphene coated Cu. **d** Schematic illustration for the evolution of defects on monolayer and bilayer graphene coated Cu. **e** AFM images of defects on monolayer and bilayer graphene coated Cu after being oxidized at 250 °C for 6 h. The detailed oxidation process is shown in Supplementary Fig. 6. **f** Time-evolution height profiles of the defects. The recording area corresponds to the dashed lines in **e**. The oxidation time (0, 0.5, 1.5, 3.5, and 6 h) is distinguished by different lightness levels.

protect Cu for more than 1000 h at 200 °C, while monolayer graphene only works for ~36 h (the breakdown limit is defined as having a 5% corroded area, Fig. 2b). This enhanced protection property of bilayer graphene was then thoroughly confirmed by oxidation experiments carried out at different temperatures (Fig. 2b and Supplementary Fig. 5). By fitting these results with the above model, the oxidation rate ($K$) of bilayer graphene is confirmed to be orders of magnitude lower than that of monolayer graphene (e.g., $4.5 \times 10^{-14} \, h^{-3}$ versus $1.2 \times 10^{-8} \, h^{-3}$ at 200 °C, see detailed comparisons in Supplementary Table 1). Then the oxidation energy barriers ($E$) were readily estimated from the Arrhenius equation[35,36] (Fig. 2c), i.e., 6.88 eV for bilayer graphene and 4.60 eV for monolayer case, confirming the significantly enhanced anticorrosion performance of bilayer graphene coating.

To have a more comprehensive understanding of oxidation behaviours at the graphene–Cu interface, we utilized a combination of in situ methods. First, we used atomic force microscopy (AFM) to track the evolution of the penetrating defects on graphene film (Fig. 2d–f and Supplementary Fig. 6). After being heated at 250 °C in air, point defects on monolayer graphene quickly expanded outward, whereas those in the bilayer graphene-coated region primarily grew in the vertical direction and formed a sharp cliff. These observations demonstrated that even if defects exist on the bilayer graphene, the interfacial diffusion of corrosive molecules at the graphene–Cu interface can be sufficiently suppressed, preventing the adjacent graphene layer from being further damaged.

Furthermore, the oxidation process under extreme working conditions was investigated by in situ environmental scanning electron microscopy (ESEM), which allowed for real-time observations with an oxygen pressure of 150 Pa, a hydrogen pressure of 80 Pa, and a temperature of 600 °C. Under such oxygen-rich and high-temperature conditions, monolayer graphene could not withstand even one second

of exposure (Fig. 3a). In contrast, the full oxidation of bilayer graphene-coated Cu took more than 120 min (Fig. 3a–f). Moreover, by tracking the corroded area, a colour-coded time-lapse image stack was obtained to depict the anticorrosion behaviour of the overlayers (Fig. 3g and Supplementary Movie 1). We can obtain that the oxidation of bilayer graphene coated Cu mainly started from the monolayer-bilayer graphene joint interface, and the expansion of the surface oxide towards the bilayer coated region was extremely slow, at a rate of only ~20 nm/min along in-plane direction.

Then, near ambient pressure X-ray photoelectron spectroscopy (NAP-XPS) was carried out to gain insight into the chemical state of Cu during oxidation. As the oxidation progressed at 150 °C, the satellite peak and Auger peak of $Cu_2O$ became increasingly prominent, proving the chemical state of Cu has risen from 0 to +1 (Fig. 3h, i). Notably, the $Cu^0$ signal persisted throughout the entire oxidation process (Fig. 3i), demonstrating that a portion of Cu has been well protected. To identify the space distribution of the oxidized area, the same sample was then directly transferred into ESEM chamber to perform the energy dispersive X-ray spectroscopy (EDS) characterization. The results showed that the $O_{K\alpha}$ peak diminished as the graphene layer number increased, indicating that Cu only survived under the protection of bilayer graphene (Fig. 3j). These findings were further supported by the time-of-flight secondary ion mass spectrometry (ToF-SIMS) analyses, where oxygen species were only detected on the monolayer coated region (Fig. 3k and Supplementary Fig. 7). By combining various in situ methods, detailed chemical state information could be integrated with visual observations in real-space at variable magnifications, confirming the effectiveness of the bilayer graphene coating strategy.

We further utilized first-principles calculations to understand the ultrahigh anticorrosion performance in the bilayer graphene/Cu

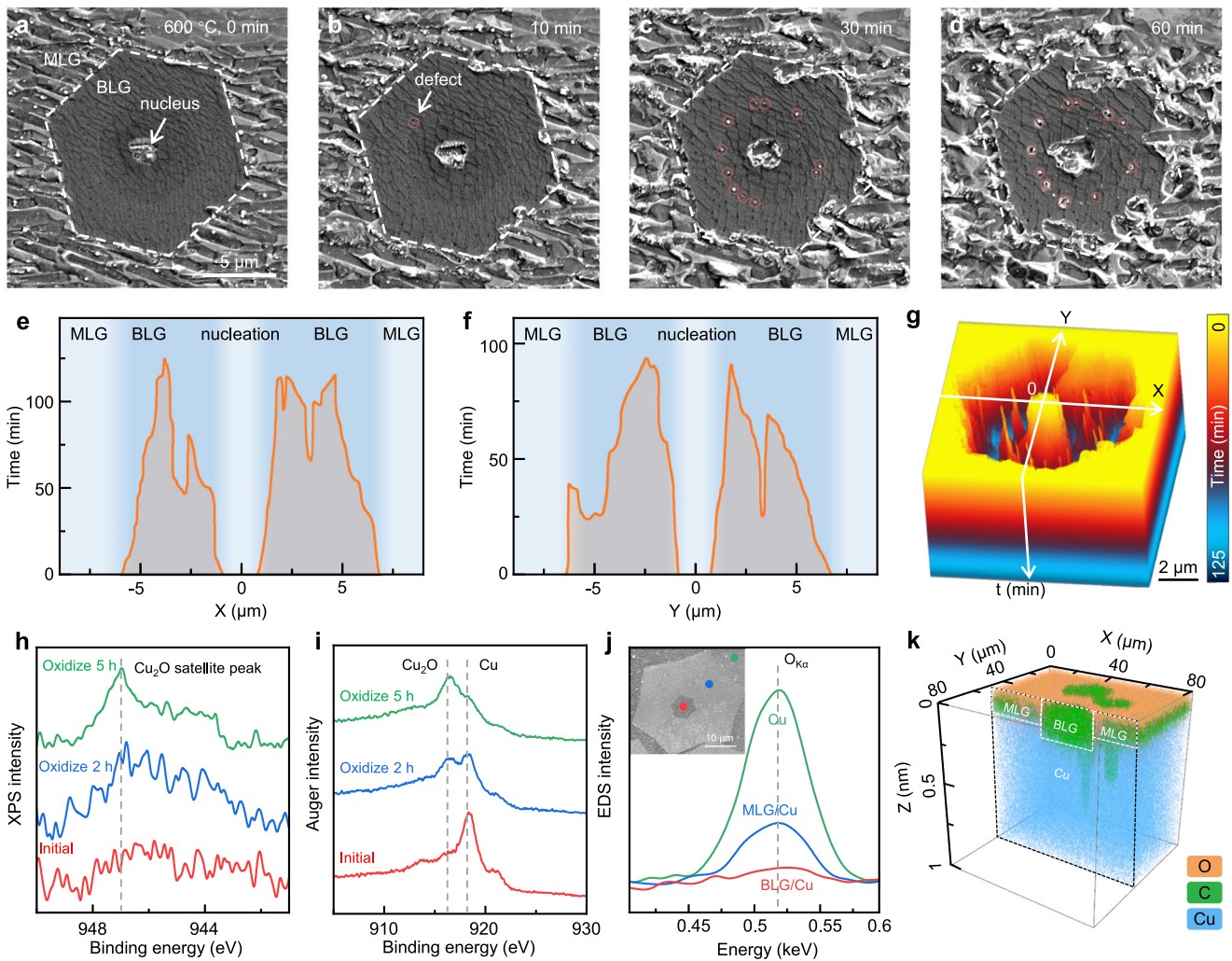

**Fig. 3 | In situ characterization of the oxidation process of Cu with graphene coating. a–d** In situ SEM images of the Cu surface coated by monolayer and bilayer graphene oxidized under 150 Pa $O_2$ and 80 Pa $H_2$ at 600 °C. Defects generated during oxidation are marked by red circles. **e–g** Time evolution plot of the sample along the X axis **e**, Y axis **f**, and its three-dimensional colour-coded stack **g**, derived from **a–d**. The non-oxidized area is shaded in orange for **e**, **f** and shown in a dented morphology for **g**. **h**, **i** In situ NAP-XPS of $Cu_2O$ satellite peak **h** and Auger peak **i** of

the Cu sample with bilayer graphene coating. The red, blue, and green curves are gathered after oxidation of 0 h, 2 h, and 5 h, respectively, in the condition of 0.3 mbar $O_2$ and 150 °C. **j** In situ EDS analysis of oxide concentration (intensity of $O_{K\alpha}$ peak) in monolayer coated (blue), bilayer coated (red), and bare (green) Cu surface after oxidation. Inset: SEM image showing the corresponding EDS probing position. **k** Cross-sectional ToF-SIMS plot of the $O^{2-}$ (orange), $C^+$ (green), and $Cu^-$ (blue) distributions in monolayer and bilayer graphene-coated regions after oxidation.

system (Supplementary Fig. 8, and see Methods for more details). The electron density difference ($\Delta\rho$) before and after graphene coating was first calculated ($\Delta\rho = \rho_{Cu+Gr} - \rho_{Cu} - \rho_{Gr}$, where $\rho_{Cu+Gr}$, $\rho_{Cu}$, and $\rho_{Gr}$ are the electron densities of graphene coated Cu, bare Cu, and free-standing graphene, respectively). The results show that the electron redistribution between Cu and the bottom layer of bilayer graphene is more prominent, indicating a strengthened interaction at the bilayer graphene-Cu interface. Moreover, there is insignificant electron density variation around the top layer graphene, demonstrating that the top layer is minimally affected by the Cu substrate and exhibits a near charge neutrality state (Fig. 4a). We have also performed the adsorption energy calculations, and confirm that bilayer graphene exhibits a stronger binding to Cu (Supplementary Fig. 9) and a reduced adsorption of corrosive molecules (Supplementary Fig. 10)[16,37,38]. Based on these findings, we propose that the bilayer graphene/Cu system demonstrates a Janus-doped characteristic, where the bottom layer is heavily doped and forms an enhanced interaction with Cu which limits the interfacial diffusion and migration of corrosive molecules, while the top layer is nearly charge neutral, thus can cut off the electron pathway and block the catastrophic electrochemical reactions. By

successfully manipulating the electronic structures of the two layers, both the interfacial diffusion and surficial electrochemical reaction pathways are suppressed, finally resulting in the ultraslow corrosion of Cu.

The proposed mechanism was verified by comprehensive electronic structure characterizations. First, the Kelvin probe force microscopy (KPFM) measurements revealed that the surface potential of bilayer graphene on Cu was ~135 mV higher than that of monolayer graphene (Fig. 4b, c), indicating a heavier electron doping effect in bilayer graphene[14,17]. Then, electronic band structure calculations and nanoscale angle-resolved photoemission spectroscopy (nano-ARPES) experiments were performed to investigate the detailed electronic structures. The calculation results demonstrate an energy shift between the Dirac point and the Fermi surface in the graphene/Cu system, and the shift of the bottom layer graphene is significantly larger than that of the monolayer (Fig. 4d for monolayer case and Fig. 4e, f for bilayer case). This shift was experimentally confirmed by nano-ARPES to be 0.27 eV for the bottom layer, and only 0.15 eV for the monolayer (Fig. 4g, h and Supplementary Fig. 11). According to the relationship between Fermi level ($E_F$) and carrier density ($n$) in

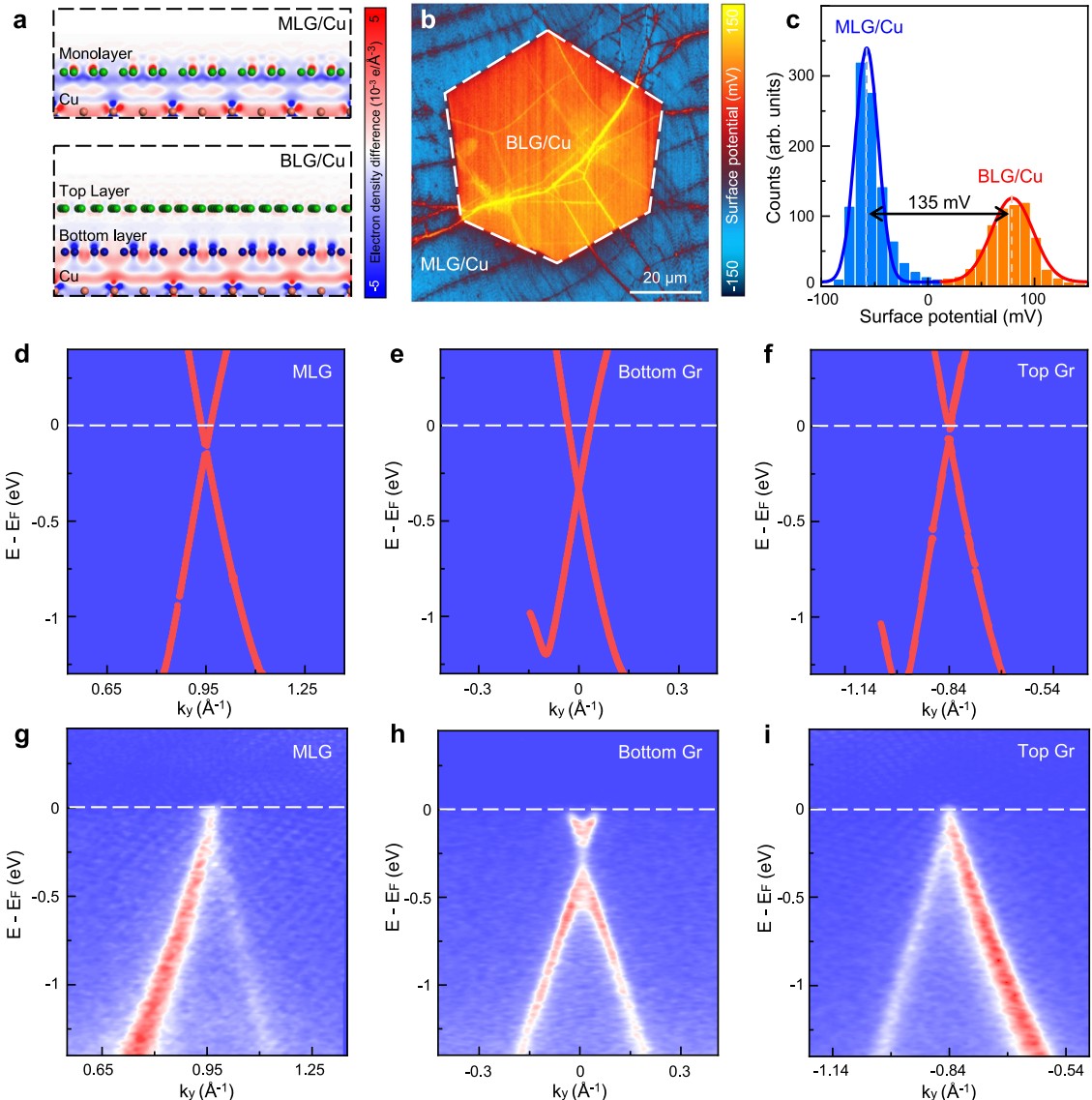

**Fig. 4 | Electronic structure characterization of Cu with graphene coating.**
**a** Calculated charge density difference of monolayer and bilayer graphene on Cu. The positive (negative) value indicates the charge density increasing (decreasing). In this figure, orange, green, and blue balls represent the Cu atom, C atom in the monolayer or top layer graphene, and C atom in the bottom layer graphene, respectively. **b** KPFM image of monolayer and bilayer graphene-coated Cu. **c** Histogram of the surface potential in **b**, showing an ~135 mV difference in the contact potential between monolayer and bilayer graphene on Cu. **d** Calculated band dispersion of monolayer graphene on Cu. **e**, **f** Calculated band dispersions of the bottom layer **e** and the top layer **f** of 27.5° twisted bilayer graphene on Cu. **g** Experimental band dispersion of monolayer graphene on Cu. **h**, **i** Experimental band dispersions of the bottom layer **h** and the top layer **i** of 27.5° twisted bilayer graphene on Cu.

graphene, $E_F(n) = \hbar|v_F|\sqrt{\pi n}$ (where $\hbar$ is the reduced Planck constant and $v_F$ is the Fermi velocity[39]), such an upshift of $E_F$ indicates a ~3 times heavier electron doping in the bottom layer (~$4.4 \times 10^{12}$ cm$^{-2}$ versus ~$1.4 \times 10^{12}$ cm$^{-2}$). This evident doping effect leads to a strengthened interaction between bottom graphene and Cu, thus effectively blocking the interfacial diffusion pathway for corrosive agents at the graphene-Cu interface (Fig. 1a).

In the case of substantial electron transfer for monolayer coating, graphene is more likely to provide electrons to corrosive agents, and its high electrical conductivity will provide a channel for electrons to migrate on the surface and participate in the cathodic reactions. Regarding the top layer of bilayer graphene, both our calculations and nano-ARPES results confirm that there are negligible electrons transferring onto the top layer (Fig. 4f, i). Such an inert top layer will hardly motivate the surficial galvanic reactions of corrosive molecules, thus efficiently reducing the electrochemical reaction rates. Therefore, the

cathodic reaction at the surface is greatly suppressed (Fig. 1a), which contributes to the enhanced anticorrosion performance of the bilayer graphene coating.

With the facile Janus-doped bilayer graphene coating design, we were able to achieve robust and effective anticorrosion performance for Cu that was three orders of magnitude better than the industrial standard recommendation (1000 h protection vs 1 h protection in air at 200 °C)[5]. As a further demonstration, we transferred bilayer graphene coatings onto Cu conductors on a printed circuit board (PCB). After annealing at 120 °C for 20 h, the bilayer graphene successfully protected the Cu conductors with no visible signs of oxidation (Supplementary Fig. 12). The remarkably improved anticorrosion ability of the atomic-thick graphene coating paves the way for the integrated applications of Cu in harsh environments, and creates significant opportunities for the miniaturisation of the next-generation electronic and optoelectronic devices.

## Methods

### Preparation of Cu with bilayer graphene coating

Bilayer graphene coated Cu samples are prepared in two ways. The first one is using the typical wet transfer method, and the bilayer coating area can be on the decimetre scale. Specifically, 8% polymethyl methacrylate (PMMA) in anisole solution was spin-coated onto the surface of the as-grown monolayer graphene/Cu sample and then the sample was baked at 120 °C for 2 min. Then the Cu was etched by 0.2 mol L$^{-1}$ $(NH_4)_2S_2O_8$ solution in a water tank. After that the obtained free-standing PMMA/graphene film was smoothly placed onto another as-grown monolayer graphene/Cu without deliberately controlling the twisted angle. Finally, the obtained PMMA/bilayer graphene/Cu sample was baked at 80 °C for 10 min and then annealed in Ar and $H_2$ atmosphere at 400 °C for 10 h. The second way is using the typical chemical vapour deposition (CVD) growth method, and these bilayer domains are of several tens of micrometres. In detail, a Cu foil was first placed on a quartz substrate and loaded into the CVD furnace. Then the system was heated to 1040 °C under a reducing atmosphere of 500 standard cubic centimetres per minute (sccm) Ar and 30 sccm $H_2$. After holding at 1040 °C for 1 hour, 0.05 sccm/50 sccm of $CH_4/H_2$ was introduced into the system for 10–30 min to obtain bilayer graphene on Cu.

### Oxidation experiments

The oxidation experiments were carried out by baking the samples in a CVD chamber in air at 120–275 °C for up to 15 days. The room temperature oxidation experiments were carried out in Beijing with the temperature controlled within 20–30 °C and the relative humidity controlled between 30 and 50%.

### Characterization

Optical images were obtained using an Olympus BX51 microscope. Raman spectra were obtained with a Witec alpha 300 R system with a laser excitation wavelength of 514 nm. EBSD characterizations were carried out using PHI 710 Scanning Auger Nanoprobe. AFM images were acquired using a Bruker Dimensional ICON system under ambient conditions. KPFM measurements were performed using Asylum Research Cypher under ambient conditions. Depth resolution mass spectra were conducted on a ToF-SIMS 5–100 instrument (ION-ToF GmbH), which relates to a UHV preparation chamber. The primary ion source uses $Bi^+$ cluster at 30 keV with an incidence angle of 45°. The dual sputtering beam uses $Cs^+$ cluster at 500 V and 27 nA with an incidence angle of 45°. The ARPES measurements were performed at the nano ARPES beamline BL07U of the Shanghai Synchrotron Radiation Facility, with a beam spot size of ~500 nm and energy/momentum resolution of 50 meV per 0.2°. Graphene sample was measured in an ultrahigh vacuum with a base pressure of more than $5 \times 10^{-11}$ mbar and photon energy of 91 eV, and the data were collected by DA30L analysers. NAP-XPS was performed on a system which is manufactured by SPECS Surface Nano Analysis GmbH. The facility is composed of three chambers, an analysis chamber, a preparation chamber, and a load-lock chamber. The X-ray source in the vacuum environment is separated from the high-pressure analysis chamber using a $Si_3N_4$ window. The analysis chamber is equipped with a PHOIBOS NAP hemispherical electron energy analyser, a microfocus monochromatized Al $K_\alpha$ X-ray source with beam size of 300 μm, a SPECS IQE-11A ion gun, and an infrared laser heater. The base pressure of the main chamber is kept below $1 \times 10^{-9}$ mbar.

### In situ observations by ESEM

The in situ experiments were conducted in a specially modified commercial ESEM (Thermo Fisher Quattro-s), using oil-free pre-vacuum pumps for the vacuum system, a homemade laser heating stage for temperature control, and a K-type thermocouple for temperature monitoring. A gas supply module facilitated by mass flow controllers (Bronkhorst) was employed, together with a mass spectrometer (Pfeiffer OmniStar) to analyse the atmosphere in the chamber. Prior to the experiments, the ESEM chamber was cleaned using plasma. The acceleration voltage was set between 3.0 and 7.5 kV, and images were captured using a large field detector. Throughout the experiments, no influence of the electron beam on the oxidation process was detected.

### Modelling of oxidation

We divide the oxidation of Cu into two processes, the first is the formation of graphene point defects, at a rate $\dot{N}$ per unit area; and the second is the defect expansion, at a rate $\dot{G}$. The expansion of defect is assumed to be isotropic here. Considering that the oxidized areas will stop expanding when they contact with each other, the fitting formula follows the Avrami equation of $Y = 1 - \exp\left(-K(t+t_0)^3\right)$ and $K = \pi\dot{N}\dot{G}^2/3$, where $Y$ is the oxidized ratio, $t_0$ is the initial oxidation level. Here, $\dot{N}$ and $\dot{G}$ follows the Arrhenius equation: $\dot{N} \propto \exp(-E_1/kT)$ and $\dot{G} \propto \exp(-E_2/kT)$, where $E_1$ is the barrier of the formation of point defects, and $E_2$ is the barrier of defects expansion. Thus, we can derive that the oxidation rate $K \propto \exp(-(E_1 + 2E_2)/kT)$. By a linear fit of $\ln K$ to $1/T$, we can derive the total barrier $E = E_1 + 2E_2$ from the slope of the fitting.

### First-principles calculation

The calculations were carried out using Vienna ab initio simulation package (VASP)[40]. The interaction between the valance electrons and ion core were described by the projector augmented wave (PAW) method[41,42]. The exchange-correlation potential was formulated by the generalised gradient approximation (GGA) with the Perdew-Burke-Ernzerhof (PBE) scheme[43]. The kinetic energy cut-off of plane wave basis is set to 417 eV. First, the bulk Cu was relaxed until the force is less than 0.02 eV/Å, and the obtained primitive lattice constant is 2.5412 Å, which is about 3% mismatch of the graphene lattice constant (2.46 Å). This relaxed Cu lattice constant was employed in the following calculations. The Cu substrate was cleaved from the bulk phase with three Cu atom layers. The bottom two layers are fixed to simulate bulk Cu, while the topmost Cu layer and bilayer graphene are relaxed until the force less than 0.02 eV/Å. As determined from the ARPES results, the bilayer graphene has ~27.5° rotation, therefore, a superlattice matrix $\begin{bmatrix} 13 & 6 \\ -6 & 7 \end{bmatrix}$ was employed to construct the top graphene. For the bottom graphene and Cu substrate, a $11 \times 11$ superlattice is applied to fit the graphene layer. For the graphene dispersion and Fermi surface simulation, the Brillouin zone unfolding was employed[44,45]. The adsorption energy was calculated by $\Delta E = E_{Sub+Ads} - E_{Sub} - E_{Ads}$, where $E_{Sub+Ads}$, $E_{Sub}$, and $E_{Ads}$ are the total energy of substrate with adsorbate, bare substrate, and adsorbate, respectively. In the calculation of corrosive molecule adsorption, one oxygen or water molecule was put on Cu. The initial top, hollow, and bridge adsorption sites were employed, and the lowest energy configurations were adopted after full relaxation.

## Data availability

The data that support the findings of this study are available within the paper and Supplementary Information. Additional data are available from the corresponding authors upon request.

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

## Acknowledgements

This work was supported by Guangdong Major Project of Basic and Applied Basic Research (2021B0301030002 (K.L.)), the National Natural Science Foundation of China (52025023 (K.L.), 51991342 (K.L.), 52021006 (K.L.), 92163206 (M.W.), 52172035 (M.W.) and T2188101 (K.L.)), the Key R&D Program of Guangdong Province (2020B010189001 (K.L.)), the Strategic Priority Research Program of Chinese Academy of Sciences (XDB33000000 (K.L.)), the Science and Technology Commission of Shanghai Municipality (22ZR1441800 (W.S.)), China Postdoctoral Science Foundation (2023M730103 (Z.B.Z.)), New Cornerstone Science Foundation through the XPLORER PRIZE (K.L.), and Double First-Class Initiative Fund of ShanghaiTech University (Z.W. and W.S.). The calculations were performed at the HPC Platform of ShanghaiTech University Library and Information Services. We acknowledge Cui Yi and Wei Wei from Suzhou Institute of Nano-Tech and Nano-Bionics for their help in the ToF-SIMS experiment.

## Author contributions

K.L., Z.W., and Z.B.Z. supervised the project and conceived the experiments. M.Z. synthesized the samples and conducted the oxidation experiments. W.S. performed the theoretical analysis. M.Z., Z.B.Z., M.D., Y.F., C.L., M.W., and X.L. performed the EBSD, Raman, and AFM characterizations. Y.L. and Z.K.L. performed the ARPES characterizations. C.X., Y.H., Z.Q.Z., and Z.L. performed the in situ SEM, TEM, EDS, and XPS characterizations. All authors discussed the results and contributed to writing the paper.

## Competing interests

The authors declare no competing interests.
