## [Peer Review File · Nature Communications]

Enhanced copper anticorrosion from Janus-doped bilayer grapheneREVIEWER COMMENTS

Reviewer #1 (Remarks to the Author):

The authors present interesting results of studies on copper (polycrystalline?) protection against corrosion by a bilayer graphene (BLG) coating.

A proposed mechanism of protection is based on two assumption:

1. Adsorption of the inner layer onto Cu
2. Chemical and charge shielding of Cu by the outer layer, i.e. it does not allow external molecules (oxygen, water etc) to reach the metal surface and does not conduct electrons vertically (Is it due to a band gap of 0.11 eV [18*]?)

The interaction of graphene with metallic surfaces was studied earlier, mainly computationally, by a couple of research groups.

It turned out that the adsorption of BLG on Cu surface leads to weak bonding (0.126 eV [18]), due to charge transfer from copper to carbon atoms (doping of graphene). The majority of the donated electrons rest in the graphene layer closest to the interface. Also some minority of donated electron charge can be seen to rest in the other graphene layer. In the case of a single graphene layer (SGL) the bonding energy was calculated to 0.033 eV [19].

The authors of the reviewed manuscript also utilized first-principles calculations, to show that the inner layer is 'heavily

doped and forms an enhanced interaction with Cu' (line 149). This should 'limit the interfacial diffusion and migration of corrosive molecules'. The following question/remarks may arise:

1. Why the authors did not calculate the adsorption energy of BGL and SGL on Cu.
2. Why the adsorption of water and oxygen was not taken into account, they can make quite strong bonds with Cu:

Water adsorption on Cu(111) = 0.12 eV [S.Ghosh et al.: Water Adsorption and Dissociation on Copper/Nickel Bimetallic Surface Alloys:Effect of Surface Temperature on Reactivity, J.Phys.Chem. C 2017,121,30,16351]

Water adsorption on polycrystalline copper = 0.47 eV [E.Colbourn et al.: Adsorption of water on polycrystalline copper: relevance to the water gas shift reaction, J.Catalysis 1991,130,514]

Oxygen adsorption on Cu(111) = 0.87 eV [16].

3. Lateral transportation of 'corrosive molecules' (i.e. between metal and protective layer) is not necessary for corrosion to proceed. If the products of corrosion reactions are soluble than they are quickly washed away and the process proceeds under the protective layer creating a cavity, which becomes deeper and wider with time. If the products of the corrosion are insoluble, e.g. oxides, then the protection locally breaks because the molar volume of oxides is larger than the one of metals.

4. The authors state rightfully that graphene can form a galvanic cell with copper. Thus water plays a crucial role in the oxidation behavior of graphene-protected Cu, and accordingly wet-oxidation needs to be studied independently. The authors carried out only one experiment of this sort without providing any details apart from duration of the experiment (Fig.1c)

line 30

'superior anticorrosion for Cu via a simple bilayer graphene coating, which provides protection for more than 5 years at room temperature...'

5. Minor remark:

line 175

'Such an inert top layer will hardly catalyse the surficial galvanic reactions of corrosive molecules...'

Comment: There are no catalytic phenomena in the galvanic processes.

* The manuscript

Reviewer #2 (Remarks to the Author):

Atomically thin anticorrosion coating, such as graphene, for Cu is an interesting topic but under debate for years. The manuscript by Zhao et al. reported the outstanding anticorrosion performance of bilayer graphene-coated Cu. The authors used a series of state-of-the-art characterization methods and did detailed experimental observations and analyses. They also proposed the operation mechanism of BLG based on simulations and further nano-ARPES verification. Overall the data and story are beautifully presented, which is impressive. In my opinion, this work is an important step to understanding the anticorrosion mechanism of graphene toward applications under some specific circumstances. I recommend publication in Nature Commun. After addressing the following comments and suggestions:

1. In Figure 1c, the color looks different at the interface region between MLG and BLG from those of MLG and BLG themselves, the authors should investigate what happened at this region and why is this.
2. Following the above question, from the proposed mechanism picture, the electrochemical reaction at the MLG/BLG interface seems important. The authors may consider if the stacking sequence, i.e., typically the second layer domain underneath the first layer film (reported in Nat. Nanotech. 2016, 11 (5), 426-431), of the BLG would affect the anticorrosion performance and the proposed mechanism.
3. A minor point: in the abstract part, "for Cu via a simple bilayer graphene coating," where "simple" may be deleted, or expressed as "simply via a bilayer graphene coating".

Reply to Referee #1

Original comment (1):

The authors present interesting results of studies on copper (polycrystalline?) protection against corrosion by a bilayer graphene (BLG) coating.

Our reply:

We sincerely thank the referee for his/her insightful suggestions and comments on our work. We found that the anticorrosion of bilayer graphene coating is effective both on single-crystal Cu (as shown in Supplementary Fig. 2) and polycrystalline Cu (as shown in Supplementary Fig. 12).

Original comment (2):

A proposed mechanism of protection is based on two assumptions:

1. Adsorption of the inner layer onto Cu,
2. Chemical and charge shielding of Cu by the outer layer, i.e., it does not allow external molecules (oxygen, water etc.) to reach the metal surface and does not conduct electrons vertically (Is it due to a band gap of 0.11 eV [18]?).

Our reply:

We totally agree with the referee that the main mechanism lies in the different roles of the two graphene layers. The bottom layer is strongly coupled to the Cu surface, thus reducing available space for interfacial diffusion. The top layer is near isolated and charge neutral, thus providing both chemical and charge shielding.

As for the origin of the charge shielding effect of the top layer graphene, as the referee kindly pointed out, the 0.11 eV band gap in the Bernal stacking bilayer graphene should play a role. In addition, in our experiments, we found that the anticorrosion performance is also effective on twisted bilayer graphene (Supplementary Fig. 3), whose band gap can be very small or even zero when the twist angle is large (Physical Review Letters 2007, 99, 256802, and the ARPES results in Fig. 4d-i in the main text). To better understand the charge shielding effect, we further carried out Bader charge calculations based on the charge density calculation (Fig. R1). The results

revealed that each carbon atom acquired ~ 0.015 electrons in the bottom layer, while acquired only ~ 0.001 electrons in the top layer. This indicates that the top layer maintains a near native state with minimal interaction with the bottom layer. As the vertical conductance is highly sensitive to the interlayer coupling (Nature Communications 2018, 9, 4068; Nano Letters 2019, 19, 3654 and Small 2020, 16, 1902844), such an isolated top layer significantly reduces electrical transport between graphene layers. Therefore, we believe that the charge shielding effect in the top layer graphene mainly stems from its weak coupling with Cu surface and its resulting charge neutral characteristic.

Fig. R1. The Bader charge calculations of bilayer graphene on Cu.

Original comment (3):

The interaction of graphene with metallic surfaces was studied earlier, mainly computationally, by a couple of research groups.

It turned out that the adsorption of BLG on Cu surface leads to weak bonding (0.126 eV [18]), due to charge transfer from copper to carbon atoms (doping of graphene). The majority of the donated electrons rest in the graphene layer closest to the interface. Also, some minority of donated electron charge can be seen to rest in the other graphene layer. In the case of a single graphene layer (SLG) the bonding energy was calculated to 0.033 eV [19].

The authors of the reviewed manuscript also utilized first-principles calculations, to show that the inner layer is 'heavily doped and forms an enhanced interaction with Cu' (line 149). This should 'limit the interfacial diffusion and migration of corrosive molecules'. The following

question/remarks may arise:

Why the authors did not calculate the adsorption energy of BLG and SLG on Cu.

Our reply:

We greatly thank the referee for the insightful suggestion on calculating the adsorption energies of graphene on Cu.

Following the referee's advice, we have calculated the adsorption energies ($\Delta E = E_{Sub+Ads} - E_{Sub} - E_{Ads}$, where $E_{Sub+Ads}$, E_{Sub} and E_{Ads} are the total energy of substrate with adsorbate, bare substrate, and adsorbate, respectively) of both monolayer and bilayer graphene on Cu (Fig. R2). The results revealed that the adsorption energy of bilayer graphene on Cu is -33.9 meV/\AA^2 , which is 3.6 meV/\AA^2 lower than that of monolayer graphene (-30.3 meV/\AA^2). This lower adsorption energy indicates an enhanced interaction strength between bilayer graphene and Cu. These computational results are consistent with part I of our proposed anticorrosion mechanism, that is, the bottom layer of bilayer graphene forms an enhanced interaction with Cu, which can effectively limit the interfacial diffusion and migration of corrosive molecules, thereby preventing oxidation.

We have added these new results in the revised manuscript.

Fig. R2. The adsorption energy of monolayer and bilayer graphene on Cu. **a**, Schematic of the calculated structure of monolayer and bilayer adsorption. **b**, The adsorption energy of monolayer and bilayer graphene on Cu. The model we used here is the same as the one we used in the main text (see the Methods section).

Original comment (4):

Why the adsorption of water and oxygen was not taken into account, they can make quite strong bonds with Cu:

Water adsorption on Cu(111) = 0.12 eV [S.Ghosh et al.: Water Adsorption and Dissociation on Copper/Nickel Bimetallic Surface Alloys: Effect of Surface Temperature on Reactivity, J. Phys. Chem. C 2017,121,30,16351]

Water adsorption on polycrystalline copper = 0.47 eV [E.Colbourn et al.: Adsorption of water on polycrystalline copper: relevance to the water gas shift reaction, J.Catalysis 1991,130,514]

Oxygen adsorption on Cu(111) = 0.87 eV [16].

Our reply:

We greatly appreciate the referee for the insightful suggestion on calculating the adsorption energies of oxygen and water. We totally agree with the referee that oxygen and water can make quite strong bonds with Cu. Therefore, Cu will be oxidized rapidly after contact with oxygen and water, as shown by the monolayer graphene coated Cu regions in the main text (Fig. 1b-c and Fig. 3a-d).

Here, following the referee's suggestion, we have performed calculations of the energies of oxygen and water adsorption on bare Cu and graphene/Cu systems (Fig. R3). We have considered the possible initial adsorption site by putting one oxygen or water molecule on the hollow, top, and bridge adsorption sites. After relaxation, the lowest energy configuration was employed to calculate the adsorption energies. Notably, for both oxygen and water molecules, the adsorption energies on bilayer graphene/Cu were larger than those on monolayer graphene/Cu. Furthermore, the adsorption energies on graphene/Cu systems are all much larger than those on bare Cu. The increased adsorption energies indicate that oxygen and water molecules are less prone to be adsorbed on bilayer graphene/Cu. These calculation results are consistent with part II of our proposed anticorrosion mechanism, that is, the top layer is close to charge neutral, which can provide a chemical and charge shielding to reduce the adsorption of oxygen and water molecules, thereby preventing oxidation.

We have added the new results in the revised manuscript.

Fig. R3. The adsorption energies of corrosive molecules on bare Cu and graphene/Cu systems. **a, c,** Schematic of the calculated structure of oxygen (a) and water (c) adsorption. **b, d,** Adsorption energy of oxygen (b) and water (d) on bare Cu, monolayer graphene/Cu, and bilayer graphene/Cu. The calculation model we used here is the same as the one we used in the main text (see the Methods section).

Original comment (5):

Lateral transportation of 'corrosive molecules' (i.e., between metal and protective layer) is not necessary for corrosion to proceed. If the products of corrosion reactions are soluble, then they are quickly washed away and the process proceeds under the protective layer creating a cavity, which becomes deeper and wider with time. If the products of the corrosion are insoluble, e.g., oxides, then the protection locally breaks because the molar volume of oxides is larger than the one of metals.

Our reply:

We greatly thank the referee for providing this insightful information on the lateral transportation process for soluble and insoluble products of corrosion. We totally agree with the referee that the horizontal transportation of corrosive molecules is not necessary for the occurrence of corrosion. Its impact becomes significant only after the corrosion has already taken place.

In the case of Cu corrosion, the resulting products (CuO or CuO₂) are insoluble, therefore they will lead to the local breakdown of the protective coating layer due to changes in molar volume. This breakdown typically occurs at regions with point defects or wrinkles and can be considered as the initiation of corrosion. Subsequently, the interfacial horizontal transport of corrosive molecules becomes increasingly significant (Advanced Materials 2018, 30, 1702944;

In the monolayer graphene coating case, the coupling between graphene and Cu is relatively weak. Consequently, corrosive molecules can continuously penetrate into the interface between the monolayer graphene and Cu from the breakdown site. This penetration further leads to the breakdown of the adjacent protective coating layer. As a result, the oxidation is found to proceed in both vertical and horizontal directions (Fig. R4a). In the bilayer graphene coating case, the coupling between graphene and Cu is stronger, making the horizontal transportation of corrosive molecules becomes much more difficult compared to the monolayer case. Consequently, the breakdown site in the bilayer graphene coating layer tends to remain localized. As a result, oxidation primarily proceeds in the vertical direction (Fig. R4b).

Therefore, the horizontal transportation of corrosive molecules is not necessary for the occurrence of corrosion, but it plays an important role in determining the corrosion rate. The inhibition of horizontal transportation is crucial for effectively suppressing corrosion.

We have updated the above discussion in the revised manuscript.

Fig. R4. Evolution of defects in the oxidation process. Time-evolution AFM images of the defects in monolayer graphene/Cu (a), and bilayer graphene/Cu (b).

Original comment (6):

The authors state rightfully that graphene can form a galvanic cell with copper. Thus, water plays a crucial role in the oxidation behaviour of graphene-protected Cu, and accordingly wet-oxidation needs to be studied independently. The authors carried out only one experiment of this sort without providing any details apart from duration of the experiment (Fig. 1c).

line 30 'superior anticorrosion for Cu via a simple bilayer graphene coating, which provides protection for more than 5 years at room temperature...'

Our reply:

We really thank the referee for this valuable suggestion on the investigation of anticorrosion performance under wet conditions. Following the referee's advice, we have conducted further oxidation experiments in the presence of water to evaluate the anticorrosion performance of bilayer graphene.

In our last manuscript, the room temperature oxidation experiment was carried out in Beijing. The temperature was controlled within 20–30 °C and the relative humidity (RH) was controlled between 30%–50%.

To simulate more severe wet-oxidation conditions, we performed a wet-oxidation test at 150 °C with an environment of 70% RH (the RH was measured at 25 °C, Fig. R5, a-c). After 4 hours of oxidation, visible cracks emerged in the monolayer graphene coated area, and gradually propagated after 40 hours of oxidation. In contrast, the region coated with bilayer graphene remained intact.

Furthermore, we immersed another sample in water and heated it to 80 °C (Fig. R5, d-f). The results showed that bilayer graphene can still effectively prevent Cu from oxidation. The monolayer graphene, however, appeared to accelerate the oxidation process. This phenomenon can be attributed to the formation of the galvanic cell facilitated by the presence of water.

We have added these new results in the revised manuscript.

Fig. R5. Wet-oxidation of Cu with monolayer and bilayer graphene coating. a-c, Time-evolution optical images of monolayer and bilayer graphene coated Cu surface oxidized at 70% RH and 150 °C. **d-f,** Time-evolution optical images of monolayer and bilayer graphene coated Cu oxidized at 80 °C in water.

Original comment (7):

Minor remark:

line 175 'Such an inert top layer will hardly catalyse the surficial galvanic reactions of corrosive molecules...'

Comment: There are no catalytic phenomena in the galvanic processes.

Our reply:

We sincerely thank the referee for pointing out this issue. We have replaced the word "catalyse" with "motivate" in the revised manuscript, and the sentence now reads as follows: "Such an inert top layer will hardly motivate the surficial galvanic reactions of corrosive molecules...".

In summary, we are very much grateful for the referee's efforts and expertise in reviewing our manuscript. These valuable comments and suggestions really help us to better understand the anticorrosion mechanism and significantly improve the quality of this work. We hope that our responses have fully addressed all the raised concerns and the referee will enjoy the revised version.

Reply to Referee #2

Original comment (1):

Atomically thin anticorrosion coating, such as graphene, for Cu is an interesting topic but under debate for years. The manuscript by Zhao et al. reported the outstanding anticorrosion performance of bilayer graphene-coated Cu. The authors used a series of state-of-the-art characterization methods and did detail experimental observations and analyses. They also proposed the operation mechanism of BLG based on simulations and further nano-ARPES verification. Overall, the data and story are beautifully presented, which is impressive. In my opinion, this work is an important step to understanding the anticorrosion mechanism of graphene toward applications under some specific circumstances. I recommend publication in Nature Commun. after addressing the following comments and suggestions:

Our reply:

We greatly appreciate the referee's positive evaluation and the recommendation of publication of our work. The raised constructive suggestions are quite helpful for us to improve the quality of this work.

Original comment (2):

In Figure 1c, the colour looks different at the interface region between MLG and BLG from those of MLG and BLG themselves, the authors should investigate what happened at this region and why is this.

Our reply:

We really thank the referee for raising this very insightful concern. This is an interesting phenomenon and warrants further investigation. Upon a careful examination of our experiments and a thorough review of the literature, we have identified the contributing factors to the higher degree of oxidation at the monolayer-bilayer joint interface.

Graphene at the interface exhibits a higher curvature due to the height difference. As a result, the adsorption energy for corrosive molecules is greatly reduced, providing additional adsorption sites (Physical Review B 2009, 79, 113409; Applied Surface Science 2019, 486, 239). As a

structural defect, the graphene edges also possess abundant dangling bonds. When exposed to air or solution, these bonds can be readily terminated by oxygen-containing groups or other reactive species. As a result, the corrosive activity of the graphene edge surpasses that of the interior graphene domain, leading to a higher oxidation rate at the monolayer-bilayer joint interface (Scientific Reports 2013, 3, 2248).

This phenomenon can be also mutually confirmed by our experimental results. For example, the *in situ* SEM experiments (Fig. R1, a-b) demonstrate that oxidation first occurs at the places where the number of layers changes (marked by red circles). Furthermore, our ToF-SIMS experiment (Fig. R1, c-d) illustrates an increase in oxygen (O^-) content at the interface, indicating the preferred occurrence of oxidation. In addition, we have also conducted adsorption experiments with selenium (Se), a congener element of oxygen, and found that its adsorption is stronger at the interface (Fig. R2). These above results confirm that the oxidation indeed tends to occur at the monolayer-bilayer joint interface.

Fig. R1. Corrosion behaviour at the interface. a, b, *In situ* SEM images of the Cu surface coated by monolayer and bilayer graphene under 230 Pa O_2 at 600 °C. c, d, Element-distribution ToF-SIMS maps of C^{2-} (a) and O^- (b), respectively.

Fig. R2. Adsorption of Se at the interface. a, SEM image of the monolayer and bilayer graphene coated Cu surface after the deposition of Se. b, EDS mapping of Se corresponding to (a).

Original comment (3):

Following the above question, from the proposed mechanism picture, the electrochemical reaction at the MLG/BLG interface seems important. The authors may consider if the stacking sequence, i.e., typically the second layer domain underneath the first layer film (reported in Nat. Nanotech. 2016, 11 (5), 426-431), of the BLG would affect the anticorrosion performance and the proposed mechanism.

Our reply:

We greatly thank the referee for raising this suggestion on the effects of the stacking sequence.

As the referee kindly pointed out, typically the second graphene layer will grow underneath the first layer (Nature Nanotechnology 2016, 11, 426), and it is indeed crucial to investigate whether the stacking sequence could impact the anticorrosion performance.

Here, we prepared the corresponding samples of different stacking sequence using the wet transfer method (Fig. R3, a and c). Subsequent corrosion experiments (Fig. R3, b and d) were carried out at 275 °C, and the results demonstrated that the stacking sequence has a minimal influence on the overall anticorrosion behaviour. Bilayer graphene continues to exhibit a superior anticorrosion performance compared to monolayer graphene, regardless of whether the second layer resides beneath or atop the first layer. We believe that this is because the stacking sequence primarily affects the configuration in the monolayer-bilayer joint interface and does not alter the electronic state of the system. Within the bilayer coating regime, the interfacial coupling between the bottom graphene and Cu remains robust, thus establishing a huge migration barrier for corrosive molecules and effectively preventing their further penetration.

Besides, there are two additional details that need to be clarified, i.e., (i) the oxidized points observed in the bilayer region (Fig. R3, b and d) may result from the defects introduced during the transfer process, and (ii) the corrosive difference in the monolayer region (a more severe corrosion is observed in Fig. R3b) can be attributed to the inferior contact between the transferred monolayer graphene with Cu.

In practical applications, it is preferred to prepare a uniform bilayer graphene film (Fig. 1b in the main text), and the hexagonal bilayer domains are presented only for demonstrative purposes. Therefore, in such cases, the interface between the monolayer and bilayer regions can be regarded

as negligible in terms of its impact on the overall anticorrosion performance.

Fig. R3. Oxidation of Cu coated by bilayer graphene with different stacking sequences. a, Schematic of the transfer process of monolayer graphene film onto as-grown graphene domains, simulating the case where the second layer grows underneath the first layer. **b,** Time-evolution optical images of the Cu surface (prepared by the method in (a)) oxidized at 275 °C. **c,** Schematic of the transfer process of graphene domains onto as-grown monolayer graphene film, simulating the case where the second layer grows atop the first layer. **d,** Time-evolution optical images of the Cu surface (prepared by the method in (c)) oxidized at 275 °C.

Original comment (4):

A minor point: in the abstract part, “for Cu via a simple bilayer graphene coating,” where “simple” may be deleted, or expressed as “simply via a bilayer graphene coating”.

Our reply:

We are very grateful for the referee on this language suggestion. We have modified the wording in the revised manuscript as “simply via a bilayer graphene coating” as the referee suggested.

In summary, we sincerely thank the referee for his/her very insightful suggestions to improve the quality of our manuscript significantly. We hope that the new results have fully addressed all the referee’s concerns and he/she will enjoy the revised version.

REVIEWERS' COMMENTS

Reviewer #1 (Remarks to the Author):

I recommend the current version of the manuscript for publication.

Reviewer #2 (Remarks to the Author):

The authors have thoroughly revised the paper according to the reviewers' comments and the quality of the paper is significantly improved. I therefore recommend publication as is.